# Modeling the Reflectance Changes Induced by Vapor Condensation in Lycaenid Butterfly Wing Scales Colored by Photonic Nanoarchitectures

**DOI:** 10.3390/nano9050759

**Published:** 2019-05-17

**Authors:** Géza I. Márk, Krisztián Kertész, Gábor Piszter, Zsolt Bálint, László P. Biró

**Affiliations:** 1Institute of Technical Physics and Materials Science, Centre for Energy Research, P.O. Box 49, H-1525 Budapest, Hungary; kertesz@mfa.kfki.hu (K.K.); piszter@mfa.kfki.hu (G.P.); biro@mfa.kfki.hu (L.P.B.); 2Hungarian Natural History Museum, Baross utca 13, H-1088 Budapest, Hungary; balint.zsolt@nhmus.hu

**Keywords:** photonic crystals, finite element calculation, capillary condensation

## Abstract

Gas/vapor sensors based on photonic band gap-type materials are attractive as they allow a quick optical readout. The photonic nanoarchitectures responsible for the coloration of the wing scales of many butterfly species possessing structural color exhibit chemical selectivity, i.e., give vapor-specific optical response signals. Modeling this complex physical-chemical process is very important to be able to exploit the possibilities of these photonic nanoarchitectures. We performed measurements of the ethanol vapor concentration-dependent reflectance spectra of the *Albulina metallica* butterfly, which exhibits structural color on both the dorsal (blue) and ventral (gold-green) wing sides. Using a numerical analysis of transmission electron microscopy (TEM) images, we revealed the details of the photonic nanoarchitecture inside the wing scales. On both sides, it is a 1D + 2D structure, a stack of layers, where the layers contain a quasi-ordered arrangement of air voids embedded in chitin. Next, we built a parametric simulation model that matched the measured spectra. The reflectance spectra were calculated by ab-initio methods by assuming variable amounts of vapor condensed to liquid in the air voids, as well as vapor concentration-dependent swelling of the chitin. From fitting the simulated results to the measured spectra, we found a similar swelling on both wing surfaces, but more liquid was found to concentrate in the smaller air voids for each vapor concentration value measured.

## 1. Introduction

The colors of butterfly wings [1,2,3,4,5,6] are generated by pigmentary (chemical) and structural (physical) factors. Both the pigments and nanostructures responsible for the color are located mostly in the wing scales. Structural colors are produced by constructive and destructive interference of the electromagnetic waves scattered on the photonic nanoarchitecture [7]. Photonic nanoarchitectures are a class of nanocomposites where light propagation obeys special rules. In the case of butterflies, their most important constituents are chitin and air. Optical properties of photonic crystal-type nanoarchitectures were first modeled by Yablonovitch [8] and John [9]. In its most general definition, a photonic crystal is a graded-refractive-index material, where the refractive index is a periodic function of the position in one-, two-, or three-dimensions, giving rise to a stop band [7]. Biological materials, however, always have a certain kind of disorder, and the disorder varies on a large scale among different butterfly species: from rigorously-ordered butterfly scales such as *Parides sesostris* [10,11] and *Cyanophrys remus* [12] to the lower end of the disorder scale as represented by seemingly amorphous structures such as those found in *Albulina metallica* [13].

A reversible change of the color and intensity occurs when the butterfly wing is subjected to vapors [14]. The first experiments were performed on *Morpho*-type butterflies [14,15,16], but vapor-dependent spectra have also been demonstrated on many other butterflies with different photonic nanoarchitectures [17,18]. The spectral changes are vapor selective, i.e., not only do they depend on the vapor concentration, but also on the kind of vapor. The selectivity was later explained, for the case of the *Morpho* butterflies, by a polarity gradient along the open Christmas-tree-like photonic structure of this butterfly species [19]. The measured change in the spectrum due to the vapors was small, but not as small as would have occurred by merely changing the contents of the air voids with the vapor, because the refractive index difference between the vapors and air was quite small [20]. One can reproduce, however, the magnitude of the color change by supposing that the vapor is partly condensed into liquid, i.e., the air voids become partly filled with liquid [20]. This phenomenon can be explained by the condensation effects, BET (Brunauer–Emmett–Teller) condensation [21], and the capillary condensation [22,23,24] process, which result in the vapor being condensed into liquid in the small voids below the saturation vapor pressure in open space. The amount of condensed vapor depends not only on the vapor concentration, but also on the size and shape of the air voids and the temperature. As additional proof of the condensation process, we demonstrated that it was possible to create a reversible color change by local cooling and warming of the wing [25]. That study also revealed that the rate of change of the color upon cooling depends on the openness of the structure, which shows that the vapor infiltrates those structures faster, where there are more channels between the air spaces, as seen in transmission electron microscopy (TEM) and scanning electron microscopy (SEM) images. *Morpho*-type butterflies [14,15,16], are known to have an open photonic nanoarchitecture, where the empty (air-filled) parts of the structure are easily accessible to the vapor. Vapor-sensitive color change was later shown to occur for the case of the closed photonic nanoarchitecture of the *Hoplia coerulea* beetle [26]. The scales of this beetle have a porous multilayer structure enclosed by a 100 nm-thick encasing envelope, termed as a “photonic cell”. In order to infiltrate the air voids, the vapor first has to penetrate the chitin walls. An important difference between the vapor-sensitive spectral response of open and closed structures is that the intensity of the main spectral peak always decreases for the open structures with increasing vapor pressure, but can increase for the case of closed structures. This was explained as an effect of the increased index of refraction of the walls between the air voids, as a result of the liquid filling of the air channels inside the walls [26].

In the present work, we report on simulations of the ethanol vapor concentration-dependent optical spectra on both the dorsal and ventral wing surfaces of the *Albulina metallica* butterfly and compare these results to measurements [27]. The *Albulina metallica* butterfly is exceptional because both of its wing surfaces possess structural color and the wing scales of both its dorsal (blue) and ventral (gold-green) sides have a similar photonic nanoarchitecture, but the characteristic sizes—distances and volumes of air voids—are different [28]. This makes it possible to study the effect of these structural parameters on the vapor-dependent spectra. Recently, we tested ten different volatiles in the 0–50% vapor concentration range on whole wing pieces of this butterfly, as well as some of the volatiles on single scales in both reflected and transmitted light [27]. Chemically-selective responses were obtained on both wing surfaces, but the spectral changes were different on the two sides. In our present work, we attempted to create a model to explain these findings. Based on the statistical analysis of the transmission electron microscopy (TEM) images, we built a simple theoretical model of the photonic nanostructure. Subsequently, the vapor-dependent optical spectrum was calculated by exactly solving the 3D Maxwell equations on the model structure utilizing the finite element method (FEM) [29]. We considered three parallel processes: (i) vapor condensation in the air voids, (ii) vapor condensation in the channels inside the chitin walls, and (iii) swelling of the structure. Chitin is known to exhibit a substance-specific swelling in various solvents [30]. First, we calculated the spectrum of a single scale, then by a simple model of the melanin absorption and the statistical distribution of scale directions, we calculated the reflectance spectrum of the whole wing.

The organization of the paper is as follows. Section 3.1, Section 3.2 and Section 3.3 contain a detailed analysis of the SEM and TEM images of the *Albulina metallica* butterfly scales. In Section 3.4, Section 3.5 and Section 3.6, we build a model based on the TEM information and calculate the vapor-dependent reflectance spectra. In Section 3.7, we compare the measured and calculated results. Section 4 is devoted to the discussion of the results.

## 2. Materials and Methods

The butterfly samples used in the present work were obtained from the collections of the Hungarian Natural History Museum, Budapest, Hungary. All of the specimens investigated were males.

### 2.1. Optical Measurements

The wings of the *Albulina metallica* males were measured using an optical spectrophotometer (Avantes AvaSpec-HS1024x122TEC, Apeldoorn, The Netherlands). For illumination, an Avantes AvaLight-DH-S-BAL balanced light source was used. The illumination and detection angles were nearly perpendicular to the wing surface, and a slight modification (<5${}^{\circ }$) was used to maximize the reflected intensity.

The vapor sensing measurements were conducted by combining a vapor mixing setup with a home-built vapor-sensing cell and the spectrophotometer [31]. The butterfly wings were mounted inside the aluminum cell, which was covered with a quartz window to provide UV transmittance. The vapor concentration was set using computer-controlled mass flow controllers that were connected to the artificial air source and the bubblers containing the test liquids. During the vapor sensing measurements, we changed the concentration of the test vapor while monitoring the spectral variations in time. We applied concentrations in the 5–50% range, with 5% steps. The details of the procedure were published recently [27].

The wavelength shift of the main visible peak was 4.5 (6.5) nm for the dorsal (ventral) side for the maximal 50% vapor concentration, and the shift was a linear function of the vapor concentration in the above-mentioned concentration range.

When examining vapor-dependent optical spectra, it is practical to introduce relative reflectance spectra. This makes it easier to visualize the small spectral changes caused by the vapors. In principle, a relative spectrum can be obtained by calculating the ratio of the spectrum with a certain vapor pressure and of the spectrum with zero vapor pressure. In practical measurements, however, relative spectra are obtained directly from the spectrometer, by letting the spectrometer store the zero vapor pressure spectrum as a reference and measuring vapor-dependent spectra relative to that [31].

### 2.2. Electron Microscopy

Scanning electron microscopy (SEM) images were taken using an LEO 1540 XB (Carl Zeiss AG, Jena, Germany) microscope on wing pieces attached with conductive tape without any preparation. The TEM samples were prepared by incorporating the wing pieces in plastic blocks, followed by ultramicrotome sectioning, which resulted in 70 nm-thick slices. The samples were examined in TECNAI 10 TEM.

### 2.3. Finite Element Calculation

Reflectance spectra were calculated by numerically solving the three-dimensional Maxwell equations by the finite element method (FEM) [29]. We applied Floquet boundary conditions in the lateral X and Y directions. The FEM calculation was performed by the Wave Optics Module [32] of the Comsol Multiphysics [33] software on a uniform wavelength grid ranging from 200–800 nm, in 2-nm steps. The refractive index of chitin and ethanol was $n_{chitin}=1.56+0.033i$ and $n_{ethanol}=1.36$ in the calculation. The small imaginary value was introduced to account for the pigment content of the scales of *Albulina metallica* [27]. The calculation was done for a normal backscattering arrangement, when both the illumination and the detection are normal to the sample surface.

## 3. Results

### 3.1. SEM and TEM Images

Figure 1 shows the top view SEM and cross-sectional TEM micrographs of a ventral wing scale of an *Albulina metallica* male. The dorsal side SEM and TEM images (not shown) were very similar to their ventral counterparts; only the characteristic sizes were somewhat smaller. A detailed comparison of the dorsal and ventral micrographs can be seen in [13,28]. The smaller length scale on the dorsal side corresponds to the color of the two wing surfaces: the dorsal wing surface of the *Albulina metallica* butterfly is blue, while the ventral wing surface is gold-green. No apparent order is seen in the arrangement of the small air voids on the top-view SEM image (Figure 1a), but as shown in [28], the two-dimensional radial distribution function (2D RDF) of the center of the air voids showed a marked first neighbor shell on both the dorsal and ventral sides, confirming the presence of a radial short-range order and the lack of angular order (a weak angular order could be seen on the ventral side). The cross-sectional TEM images (Figure 1b) showed, however, a layered structure; there were 3–5 pairs of layers where layer “A” is a perforated chitin sheet containing air voids and layer “B” is apparently solid chitin. The scale shown in Figure 1b has four “B” layers and three “A” layers; see Figure 2 for notation. This means that the 3D photonic nanoarchitecture of *Albulina metallica* is a 1D + 2D structure, where the vertical structure is a finite multilayer, i.e., a 1D photonic crystal (a “1D photonic cluster”, meaning a periodic arrangement of a few layers), and the lateral structure is a quasi-random arrangement of air voids possessing a radial short-range order. According to our measurements on the SEM and TEM images [28], the vertical layer periodicity $p_{ab}=d_{a}+d_{b}$ and the average lateral distance of the air voids had a similar value on both sides, and it was 200 nm and 260 nm on the dorsal and ventral sides, respectively. $d_{a}$ ($d_{b}$) denotes the thickness of the “A” (“B”) layers, respectively.

The top-view SEM image (Figure 1a) shows that the topmost chitin layer (“B” layer) was porous, as we could “see into” the air voids of the topmost “A” layer (the pores are the darker spots on the SEM image). The size of the holes changed randomly; in the case of the larger holes, we could see through the “A” layer and see the second “B” layer, which was also porous. In the case of the TEM micrograph (Figure 1b), we could also see discontinuities in the chitin layers (the “B” layers) separating the perforated layers (the “A” layers), and also when the chitin layer seemed to be continuous, its gray level was not constant. Note that the TEM image is a cross-sectional image, but the slice is not infinitely thin; the TEM image is a kind of projection image through a 70 nm-thick slice, and the 70-nm thickness is in the same range as the size of the smaller pores. Hence, if this slice incorporated a part of an air void, the corresponding pixels became a lighter gray in the TEM image. These findings show that there are vertical channels present in this structural element, i.e., those connecting air voids in adjacent “A” layers through a “B” layer. Similar SEM and TEM images were seen on the scale structures of several polyommatine butterflies [34]. The TEM image also revealed the presence of lateral channels (those connecting adjacent spheres within the same “A” layer) because some of the air voids could be seen to be connected.

In the next subsection, we more carefully analyze the TEM image and thus build an appropriate model structure of the photonic nanoarchitecture filling the scales of the *Albulina metallica* butterfly.

### 3.2. Direct Space Averaging Method

Previously, we developed a direct space averaging (DSA) algorithm [28] in order to study the order-disorder effects in butterfly wing scales. The method was based on averaging the local environments of the repetitive units. When applied to the top-view SEM images of different butterflies, it was able to differentiate structures with a long range-, medium range-, and short-range order. In the case of the SEM images, the DSA analysis involved three consecutive steps: (i) find the middle of the holes; (ii) cut out a rectangular sub-image centered on each hole; and (iii) calculate an averaged grayscale image from these sub-images.

In this section, we further developed the DSA method in order to study the cross-sectional TEM images. As already noted in Section 3.1, the gray level of the “B” layers was not homogeneous in the TEM image, which indicates the presence of vertical air channels crossing the chitin layers. We wanted to determine whether there was a correlation between the position of the vertical air channels in the “B” layers and the position of the air voids in the “A” layer. In our original DSA algorithm, we simply cut out sub-images centered on each hole and averaged them. Now, we intended to do the same for the air voids of the middle “A” layer of the TEM image ($A_{2}$ layer in Figure 2a), but we had to compensate for two additional factors: (i) the layers were not perfectly flat; and (ii) the first neighbor distances of the adjacent air voids varied. Now, we intended to “transform out” both of these factors because we wanted to study not only the middle “A” layer itself, but also the relative position of the air spaces between the different layers, which include: (i) air voids in the “A” layers, (ii) horizontal channels in the “A” layers between the air voids, and (iii) vertical channels in the “B” layers.

Hence, the enhanced algorithm of the DSA method is as follows:FOR each hole in the middle “A” layer ($A_{2}$ layer)
-FIND the center of the hole: ${\vec{r}}_{hole}$ and the center of its left and right neighbors: ${\vec{r}}_{left}$, ${\vec{r}}_{right}$.-CALCULATE the length and angle of the line section connecting the left and right neighbors, $d=∥{\vec{r}}_{right}-{\vec{r}}_{left}∥$, $\alpha =\mathrm{angle}\left({\vec{r}}_{right},{\vec{r}}_{left}\right)$.-SHIFT the image with ${\vec{r}}_{shift}=-{\vec{r}}_{hole}$, so that the present void is shifted to the center of the image.-ROTATE the image with −α, so that the line section connecting the left and right neighbors are horizontal.-SCALE (enlarge or shrink) the image so that the length *d* (distance of left and right neighbors) are the same on each image.AVERAGE gray values of all shifted images (pixel-by-pixel).

Thus, we cut out sub-images centered on the holes, rotated and scaled them, and then averaged them. The technical details of the procedure are the same as those in [28]. The centers of the holes are shown by red dots in Figure 2a. The leftmost and rightmost dots were excluded from the averaging because each void needed to have a left and right neighbor in the algorithm.

### 3.3. Results of TEM DSA Analysis

The resulting DSA image is shown in Figure 2b and its linecuts in Figure 2c. The DSA image is oriented in the same way as the original TEM micrograph (Figure 2a), the wing surface facing down, so this orientation symbolizes that it is a ventral scale. Its most prominent features are the horizontally-aligned three large white spots in the middle. The central white spot is the average central air void, and the other two white spots are its left and right neighbors. The two light gray horizontal bands on the bottom and top of the image are the average of the two “A” layers ($A_{1}$ and $A_{3}$; cf. Figure 2b) adjacent to the middle “A” layer ($A_{2}$). The gray level of these bands was nearly homogeneous, though some structure was present. There was noise due to the small number of averaged sub-images. While in the case of typical SEM images, there were several hundred holes, we only had 36 air voids in the TEM image in Figure 2a. This is because we need special, smaller magnification, but high resolution TEM images in order to produce a better statistics. Such measurements are currently underway. A weak periodic gray level variation was seen in the upper band ($A_{3}$), and we highlighted the maxima in Figure 2b with the dotted circles. These circles show that there was a weak correlation between the air void positions in the middle and upper “A” layers ($A_{2}$ and $A_{3}$), and the air voids in the upper layer tended to be located at sites above the walls separating the voids in the middle “A” layer ($A_{2}$).

The red and blue horizontal lines in Figure 2b show the positions of the two averaged “B” layers ($B_{2}$ and $B_{3}$), adjacent to the middle “A” layer ($A_{2}$). The red and blue linecuts, shown in Figure 2c, were taken along these lines. These linecuts show a marked quasi-periodicity, which indicates that there were air channels in the “B” layers and that the position of these air channels in the “B” layers was not random relative to the air voids in the “A” layers, but there was some correlation between the position of the air spaces in the adjacent “A” and “B” layers.

These findings led us to build the model structure; see Section 3.4 for details.

### 3.4. Building the 3D Model of the Scale

The TEM image analysis shown in the previous section (Section 3.3) indicates that the wing scales of the *Albulina metallica* butterfly had a 1D + 2D structure, meaning that the photonic nanoarchitecture is composed of A-B layer pairs, and in fact, both of the layers themselves have a porous structure; only the amount of porosity is different in the “A” and “B” layers. Based on the fact that most of the air voids in the “A” layers were completely white in the TEM images, we can conclude that the diameter of these air voids was larger than the TEM slice width (70 nm). Previously, we analyzed TEM slices cut in different directions [28] where the cutting planes were always perpendicular to the scale surface and rotated around a line perpendicular to the scale surface. That analysis showed that the air voids were spherical and had an isotropic arrangement inside the “A” layers. TEM DSA analysis (Section 3.3) indicated the presence of horizontal and vertical channels connecting these spherical airspaces. The average diameter of those channels was smaller than the diameter of the air voids, because the channels appeared gray in the TEM and TEM DSA images (cf. Figure 1 and Figure 2b).

The model structure (Figure 3) utilized in this paper was constructed based on the findings summarized in the previous paragraph. First, we measured the average sizes and distances of the air voids in the “A” layers, as well as the $d_{a}$ and $d_{b}$ thicknesses of the “A” and “B” layers in the SEM and TEM images, then we inserted horizontal and vertical air channels. The diameter of these air channels was determined by calculating the main visible reflectance spectral peak wavelength with the algorithm given in [35] and fitting its value to the measured peak position by varying the diameter of the air channels as a parameter. This calculation was based on the so-called perforation factor. The perforation factor *P* is the ratio of the total volume of the air voids in the structure and the total volume of the structure, $P=V_{air}/V_{total}=V_{air}/\left(V_{air}+V_{chitin}\right).$ With this, the effective refractive index is $n_{eff}=Pn_{air}+\left(1-P\right)n_{chitin}$. We performed this analysis for the “A” and “B” layers separately and thus obtained the effective refractive indices of the two layers, $n_{a}$ and $n_{b}$. The main visible reflectance spectral peak wavelength [7] is given as $\lambda_{main}=2p_{eff}$, where $p_{eff}=n_{a}d_{a}+n_{b}d_{b}$ is the vertical effective layer periodicity.

Our geometry (Figure 3) was similar to the one we constructed earlier [36], where the “A” layers contained air voids and chitin “B” layers separated the “A” layers, but this time, we also inserted horizontal and vertical air channels. All air spaces were supposed to have a cuboid shape, though the real air voids and channels are spherical and helical, i.e., bounded by curved surfaces. This approximation makes the FEM spectrum calculations less demanding. We performed, however, a few test calculations with spherical air voids and tube-shaped channels constructed so that their volume was equal to our cuboid air voids and channels. The optical spectrum was practically unchanged in the visible range; however, considerable changes were seen in the UV range. Since in this paper, our main focus was on the visible part of the spectrum, the cuboid approximation was sufficient. It is not an easy task to demonstrate the 3D structure, because the air voids are inside the chitin. Hence, Figure 3 only displays the air spaces, rendered semi-transparent, without the chitin. The sizes of the model structure are given in the vertical and horizontal cross-sections shown in Figure 4 and in Table 1, where $d_{a}$ and $d_{b}$ are the thickness of the “A” and “B” layers; $p_{z}=d_{a}+d_{b}$ is the vertical periodicity (height of one story); $p_{xy}$ is the lateral periodicity; $w_{a}$ and $w_{b}$ are the thicknesses of the horizontal and vertical channels; and *c* is the length of the horizontal channel. The length of the vertical channel was identical to the “B” layer thickness, $d_{b}$. The $\left\{v_{x},v_{y},v_{z}\right\}$ dimensions of the air voids could be easily derived as $v_{x}=v_{y}=v_{xy}=p_{xy}-c$, $v_{z}=d_{a}$.

Another major approximation we applied here was that our model structure was perfectly ordered, even though the real biological structure is always disordered. We address the validity and the ramifications of this approximation in Section 4, but the main conclusion is that for a normal backscattered configuration, the spectrum was unchanged in the first order when we introduced a lateral randomness into the positions of the air voids.

### 3.5. Three-Dimensional Finite Element Calculation of the Vapor-Dependent Reflectance Spectrum of a Scale 

Reflectance spectra for the normal backscattered configuration were calculated by numerically solving the three-dimensional Maxwell equations by the finite element method; see Section 2.3 for details. The resulting reflectance curve is shown in Figure 5 by a solid line.

When the butterfly wing is subjected to vapors, liquid is condensed in the air voids. By increasing the vapor concentration, the amount of the condensed liquid also increases. The amount of condensed liquid also depends on the size of the air spaces, as the smaller air spaces are filled first. Therefore, according to the investigation of the SEM and TEM images in Section 3.1 and Section 3.2, the air spaces in the “B” layers are smaller than the air spaces in the “A” layers, so we assumed that with an increasing concentration, the “B” layers are filled first before the “A” layers. Hence, if the filling factor (ratio of liquid filled volume to the total air volume) is $f_{A}$ for the “A” layer and $f_{B}$ for the “B” layer, then we can introduce, with a linear transformation, the constants $c_{B}$ and $c_{AB}$, where $f_{A}=c_{AB}$ and $f_{B}=c_{AB}+c_{B}$. Here, $c_{B}$ corresponds to the process when the air spaces are filled in the “B” layer only, and $c_{AB}$ corresponds to the process when the air spaces are filled in both the “A” and “B” layers. The third theoretical possibility, when only the “A” layers are filled and the “B” layers are empty, does not have a physical significance because the voids in the “A” layers are larger than those in the “B” layers; hence, the voids in the “B” layers are filled first.

When the wing is exposed to vapor, the liquid does not condense evenly in all air voids [37]; for a given concentration, when the smaller voids are already filled, the larger ones may be still empty, or only partially filled. Given that for small liquid concentrations, the average lateral distance of the liquid filled voids is larger than the coherence length [38] of white light, the reflectance components from the air-filled and from the liquid-filled voids are added incoherently. Hence, in our model of the vapor-dependent spectra, we calculated the reflectance of the partially liquid-filled nanoarchitecture as a weighted average of the spectra calculated for completely air-filled and completely liquid-filled configurations. First, we calculated the reflectance by assuming that all of the air spaces were filled with air, which is the $R_{0}\left(\lambda \right)$ spectrum (solid curve in Figure 5). Then, we repeated the calculation for the B filling process, which is the $R_{B}\left(\lambda \right)$ curve (dashed curve in Figure 5), and for the AB filling process, which is the $R_{AB}\left(\lambda \right)$ curve (dotted curve in Figure 5).

We also accounted for the swelling effect of the chitin matrix [30] in our calculation. Assuming an isotropic enlargement of the photonic nanoarchitecture by a factor of *s*, the swelling shifts the spectrum to larger wavelengths.

Summing up, the reflectance spectrum is given by:(1)$$R\left(\lambda ;c_{B},c_{AB},s\right)=\left(1-c_{B}-c_{AB}\right)R_{0}\left(\lambda /s\right)+c_{B}R_{B}\left(\lambda /s\right)+c_{AB}R_{AB}\left(\lambda /s\right).$$

### 3.6. Reflectance Spectrum of the Whole Wing 

So far, the calculation provided the normal reflectance spectrum of one scale. In order to approximate the reflectance spectrum of a whole wing, two other important factors have to be accounted for: (i) the effect of the cover and ground scales and (ii) the effect of the random angular distribution of the scales.

(i) Lycaenid butterflies almost always contain absorbing pigments, mainly melanin [39]. In most cases, the butterfly wing contains several layers of scales [40], and the melanin is mainly concentrated in the ground scales [5]. Moreover, melanin can be eumelanin or pheomelanin, which have different absorption spectra [41]. Hence, in order to create a realistic optical model of the butterfly wing, we had to insert the wavelength-dependent absorption of the melanin [41,42,43] into the model. In our approximation, this was done by applying a linear approximation. Since we have no direct experimental data about the kind and distribution of melanin in this photonic nanoarchitecture, we relied on the experimental reflectance spectra. At the zeroth level, we added a small constant imaginary part to the refractive index of chitin for the scales containing the photonic nanoarchitecture. Assuming a wavelength-independent absorption is an oversimplification; hence, at the first level of approximation, we added a wavelength-dependent linear background to the reflectance spectrum calculated in Section 3.5. This calculation is the reverse of that applied [34] when analyzing the experimental whole wing reflectance spectra. The intercept and slope of the linear background line was obtained by fitting a linear function to the small- and large-wavelength parts of the experimental whole wing reflectance spectra.

(ii) The random angular distribution of the scales was accounted for by applying a Gaussian broadening to the spectrum.

Figure 6 shows the effects of the three simulation parameters (*s*, $c_{B}$, and $c_{AB}$) on the reflectance spectra. In order to reveal the effect of the parameters, we selected large parameter values, which do not occur in practical butterfly spectra. The insets show the spectra with the broadening already applied, but without the background correction (the background line is also shown in the insets). The left and right sub-figures show the direct and relative reflectance spectra, respectively.

Figure 6a,b shows the effect of the *s* parameter, the swelling. The isotropic enlargement of the structure causes a red shift, R(λ)→R(sλ). The intensity of the peaks is unchanged (cf. Figure 6a inset), but after adding the background line, the intensity of the larger wavelength peaks also became larger (cf. Figure 6a). The red shift caused the appearance of a pair of negative and positive peaks in the relative spectra (cf. Figure 6b) on the left and right sides of the main peak wavelength. The amplitude of the negative peak was somewhat smaller, due to the background correction.Figure 6c,d shows the effect of the $c_{B}$ parameter. The intensity of the reflectance increased with an increase in $c_{B}$ because the ethanol filling increased the effective index of refraction of the “B” layer, and this increased the refractive index contrast of the “A” and “B” layers. The wavelength of the peak also increased, because an increase of the index of refraction increased the optical thickness of the structure. The interplay of these effects resulted in the left side of the reflectance curve being practically unchanged, but there was an increase in reflectance at the peak region and on the right side. This caused the appearance of a large positive peak in the relative reflectance spectrum (cf. Figure 6d) at the right side of the main peak wavelength.Figure 6e,f shows the effect of the $c_{AB}$ parameter. The main effect was the decrease of the reflectance, caused by the decrease of the refractive index contrast between the two layers. This in turn caused a decrease in the value of the relative reflectance (cf. Figure 6f) at the main peak wavelength with increasing *c*.

### 3.7. Comparison of the Measured and Simulated Reflectance Spectra

Figure 7a,b shows the measured dorsal and ventral relative reflectance spectra for increasing vapor concentration. Since the vapors only caused small changes in the spectra, especially for small concentrations, only the relative spectra are displayed. The dorsal and ventral spectra, apart from a red shift caused by the larger length scale of the ventral photonic nanoarchitecture (cf. Table 1) were similar: there was a negative (positive) peak at the left, bluer (right, redder) side of the main visual spectrum peak. The amplitude of the positive peak (at the redder side) was larger than the negative peak (that at the bluer side), however only in the case of the ventral spectrum. This indicates that the change of the spectrum was larger at the right, redder side of the main peak for the ventral side.

Next, we reproduced these measured relative reflectance spectra by fitting the three free parameters, *s*, $c_{B}$, and $c_{AB}$ of the theoretical Equation (Equation 1) derived in Section 3.5, and we also applied the linear background (effect of the ground scales) and broadening (effect of the disorder) (Section 3.6). The fitting was performed by a least-squares fit by minimizing the integral of the square of the residuals:(2)$$\Delta \left(x;c_{B},c_{AB},s\right)=\int_{\lambda_{min}}^{\lambda_{max}}{\left[R_{measured}\left(\lambda ;x;c_{B},c_{AB},s\right)-R_{calculated}\left(\lambda ;x;c_{B},c_{AB},s\right)\right]}^{2}d\lambda ,$$
where “*x*” is the ethanol vapor concentration; $\lambda_{min}=$ 200 nm; and $\lambda_{max}=$ 800 nm. The residuals are linear in $c_{B}$ and $c_{AB}$, but nonlinear in “*s*”. For simplicity, we performed the fitting procedure only for the largest, 50% ethanol vapor concentration (x= 0.5) and obtained the values of the fitting parameters for the smaller ethanol concentrations by assuming that they depended linearly on “*x*”. Figure 7c,d shows the calculated spectra for increasing “*x*” parameter values. It is perfectly possible to perform the fitting for each ethanol concentration separately, but as seen in Figure 7, the linear approximation sufficiently fits the measurements for these small concentrations.

The fitted parameters for the dorsal side were: $c_{B}=0.08x,c_{AB}=0.08x$, and s=0.016x, where “*x*” is the ethanol vapor concentration, which means that $c_{B}=0.04,c_{AB}=0.04$, and s=0.008 for the largest measured vapor concentration x=0.5. By calculating the $f_{A}=c_{AB}$ and $f_{B}=c_{AB}+c_{B}$ filling factors of the “A” and “B” layers (cf. Section 3.5), we obtained that for a x=0.5 vapor concentration, there was a $f_{B}=$ 8% liquid ethanol filling of the air voids in the “B” layer and $f_{A}=$ 4% filling in the “A” layer, besides the 0.8% swelling.The fitted parameters for the ventral side were: $c_{B}=0.4x,c_{AB}=0$, and s=0.012x, which means that $c_{B}=0.2,c_{AB}=0$, and s=0.006 for the largest measured concentration x=0.5. Thus, we obtained that for a x=0.5 vapor concentration, there was $f_{B}=$ 20% liquid ethanol filling of the air voids in the “B” layer and $f_{A}=$ 0% filling in the “A” layer, besides the 0.6% swelling.

## 4. Discussion

As we learned from the examination of the SEM and TEM images of the *Albulina metallica* butterfly wings (cf. Section 3.1 and Section 3.3 and [13,28]), the photonic nanoarchitecture filling the wing scales had a 1D + 2D structure, i.e., it was composed of pairs of nearly flat chitin layers, and the layers themselves contained a quasi-random arrangement of air voids, while adjacent chitin layers were separated by layers where the principal component was air, with a small fraction of chitin (cf. Figure 1).

Next, we examined the effect of the lateral disorder on the normal backscattered spectrum. Previously, we had performed goniometric spectrum calculations [28] for ordered and disordered lattices of scattering centers by utilizing the first Born approximation [44,45]. The results showed that in this approximation, the normal backscattered spectrum did not change by introducing a lateral disorder. The disorder only affected the angular dependence of the spectrum. This enabled us to model the photonic nanoarchitecture as an ordered model because we only analyzed the normal backscattered spectra in this paper.

Our model had three open parameters, $c_{AB}$, $c_{B}$, and *s* (the swelling factor), and the filling factors of the “A” and “B” layers depended on the *c* parameters linearly, $f_{A}=c_{AB}$ and $f_{B}=c_{AB}+c_{B}$. When the butterfly wing was exposed to a vapor of a specific substance with a specific vapor concentration and temperature, the detailed physical and chemical [19] interaction processes between the vapor and the photonic nanoarchitecture determined the specific values of the three parameters. As shown in [31], these physical and chemical processes are selective to the kind of the vapor; thus, the determination of the values of the $c_{B}\left(x\right)$, $c_{AB}\left(x\right)$, and s(x) functions (where *x* is the vapor concentration) helps to understand these microscopic processes. The different swelling factors associated with different substances [30] have an important contribution to the chemical selectivity. In the case of the experiments with ethanol for the *Albulina metallica* butterfly, the swelling was similar on the dorsal and ventral sides, but we saw a marked difference in the filling processes: the amount of the condensed liquid was larger at the ventral side, but it was confined only to the “B” layers, while both layers were filled on the dorsal side, though the filling of the “B” layers were somewhat larger. The negligible filling of the ventral “A” layers can be attributed to the large size of the air voids. The larger filling of the ventral “B” layers implies that the air channels in the “B” layers may be composed of several smaller diameter channels. The larger measured main peak shift on the ventral side (6.5 nm, as compared to 4.5 nm at the dorsal side) for the x=0.5 vapor concentration can be easily explained by the larger vertical effective layer periodicity $p_{eff}$ of the ventral scale structure. Indeed, the ratios of the peak shifted to the geometric thickness, 4.5 nm/200 nm = 0.0225 at the dorsal side and 6.5 nm/260 nm = 0.025 at the ventral side had a similar value, which shows that the larger peak shift on the ventral side was due to the larger size of the ventral structure.

A simplified model of the butterfly wing can be composed of the following structural elements: (i) cover scales (on the dorsal side); (ii) ground scales (on the dorsal side); (iii) wing membrane; (iv) ground scales (on the ventral side); and (v) cover scales (on the ventral side). The thickness of these objects are in the 1 μm range, and there are air gaps separating them. The distance between the above-mentioned structural elements is also in the μm range; hence, we did not expect coherent scattering between them, as coherent scattering may occur only inside the structural elements. Such incoherent scattering color calculations were performed for the case of vertebrate color patches [46], which are 3D structures that often contain multiple pigment types and structural features. In lycaenid butterfly wings, most of the melanin [47] is concentrated in the ground scales; therefore, considerable absorption only occurs there. In this paper, we modeled the melanin absorption by adding a linear background to the FEM calculated spectrum.

In the present paper, we only measured the normal backscattered spectrum and focused on the visible part of the spectrum. The fine details of the structure (e.g., the precise shapes and dimensions of the air voids) can, in principle, be revealed by measuring and analyzing angle-dependent spectra in the UV range. Such studies are currently underway.

## 5. Conclusions

Chemically-selective vapor sensors with an optical readout can be realized with the photonic nanoarchitectures occurring in butterfly wing scales [17,26] and by bioinspired sensors [48,49,50]. The *Albulina metallica* butterfly is exceptional because the wing scales of both its dorsal (blue) and ventral (gold-green) sides have a similar [28] 1D + 2D photonic nanoarchitecture.

We performed ethanol vapor-dependent relative reflectance spectrum measurements [27] on both the dorsal and ventral sides and analyzed the change of reflectance due to the vapor in the visible range. As we have recently shown [27] by principal component analysis (PCA), the structural colors of the two sides have different vapor sensing properties, and both the intensity and the wavelength can change. The change (increase) of the wavelength is larger on the ventral side, which can be explained by the larger air voids on the ventral side. The change of the intensity is different on the two sides because the tendency of the refractive index ratio is different.

In order to simulate these vapor-dependent relative reflectance spectra, we built a model, which was composed of four “A”-“B” layer pairs (corresponding to the number of layers observed in the TEM images). The “A” layers contain large air voids, which are interconnected by air channels, horizontal channels in the “A” layers and vertical channels crossing the “B” layers; these vertical channels are the air spaces in the “B” layers. When exposed to vapor, liquid is concentrated into the air voids, and the amount of concentrated liquid depends on the size and shape of the voids, as well as on the accessibility of the pores and the partial pressure. Therefore, the average liquid content of the “A” and “B” layers will be different for a given vapor concentration due to the different sizes of the air voids in the two layers. Next, we calculated the reflectance spectrum by exactly solving the 3D Maxwell equations with the help of a parametric FEM calculation. We found that the swelling of the nanoarchitecture was similar on the two wing sides, but the liquid filling of the “A” and “B” layers was different on the dorsal and ventral sides. This can be attributed to the different sizes and shapes of the air voids on the two sides.

It is worth pointing out that, to our knowledge, the qualitative agreement between the measured and computed response curves, when using lycaenid butterfly wings as chemical sensors, was achieved for the first time. The same structural model was able to reproduce the behavior of two distinct structures, one of blue and one of gold-green color, occurring on the wings of the same butterfly species. This is an indication that this type of model may be suitable for describing the behavior of numerous other butterfly scales of lycaenid butterflies possessing the so-called “pepper-pot”-type nanoarchitectures.

## Figures and Tables

**Figure 1 nanomaterials-09-00759-f001:**
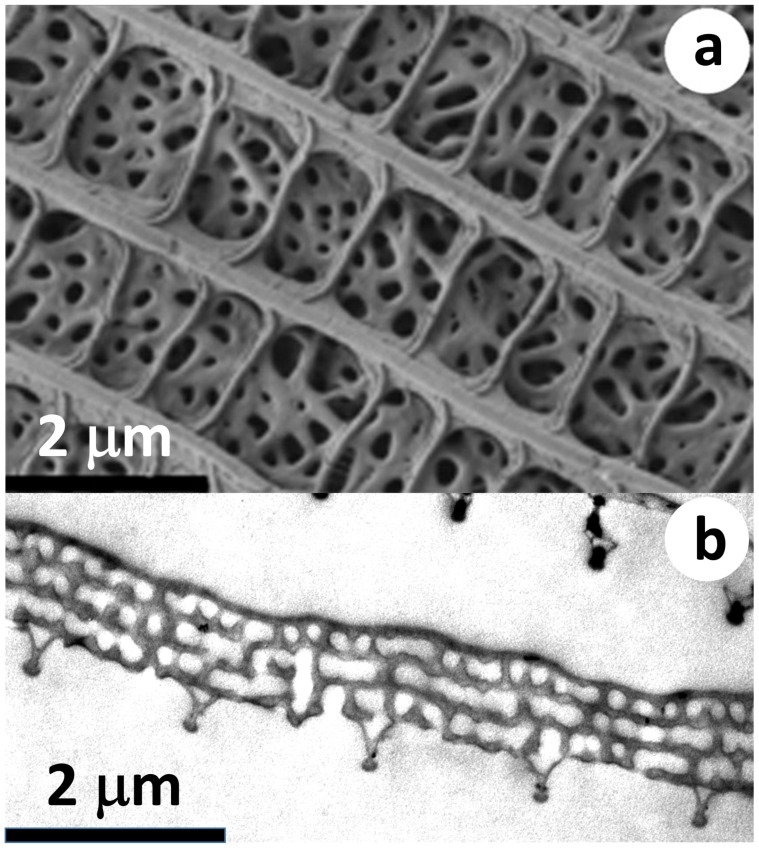
Electron micrographs of the scales on the ventral wing surface of an *Albulina metallica* male. (**a**) Top view SEM micrograph of a cover scale (from [13]); the dark holes correspond to the air voids. (**b**) Cross-sectional TEM micrograph of a cover scale (from [28]). The light gray regions correspond to the air voids. The five triangular downward protrusions are the cross-sections of the ridges. The scale is displayed facing down, i.e., the ridges are pointing downwards to symbolize that it is a ventral scale. Note the edge of another scale in the upper right corner of the micrograph, which is a ground scale. The SEM and TEM images do not show the same scales.

**Figure 2 nanomaterials-09-00759-f002:**
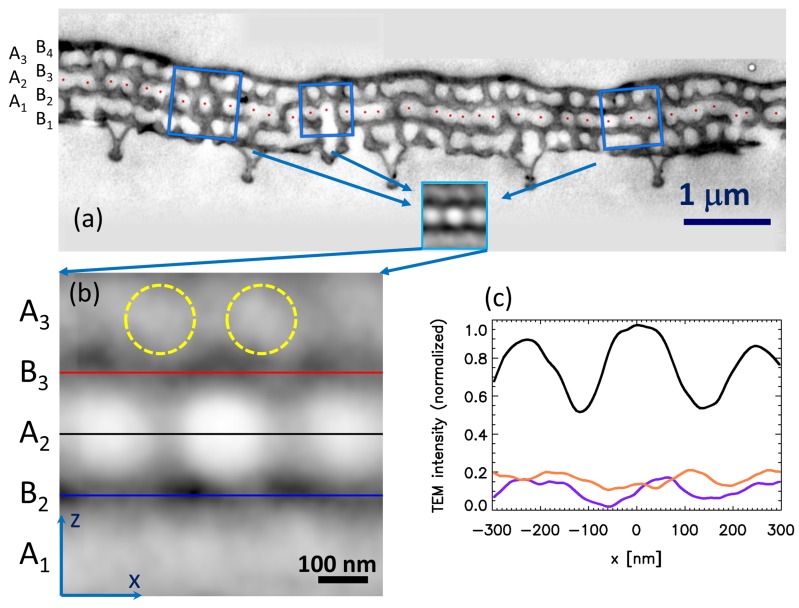
Direct space averaging applied to a TEM micrograph. (**a**) Cross-sectional TEM image of a ventral cover scale of *Albulina metallica* with four chitin layers (“B” layers) and three perforated chitin layers containing spherical air voids (“A” layers). The scale is shown facing down to illustrate that this is a ventral scale. The red dots show the centers of the air voids in the middle perforated layer ($A_{2}$). The three blue squares illustrate the averaging process for three selected air voids, but in the real calculation, the averaging was done for all of the air voids (see the text for details). The inset shows the calculated averaged image. (**b**) DSA processed image of the TEM micrograph shown in (a) (same as the inset in (a), but enlarged). The horizontal lines show the positions where linecuts were calculated (in the $B_{2}$ and $B_{3}$ layers). The yellow dashed circles show the positions of the intensity maxima in the $A_{3}$ layer. *x* (*z*) is the horizontal (vertical) direction, respectively. (**c**) Linecuts from the DSA image in (b). The black line shows the grayscale intensity of image (b) along the middle perforated layer ($A_{2}$ layer), and the red and blue lines show the intensity along the two adjacent chitin layers ($B_{2}$ and $B_{3}$ layers). The intensity value is scaled into the [0,1] interval, and zero means the minimum intensity, one the maximum intensity (see the text for details).

**Figure 3 nanomaterials-09-00759-f003:**
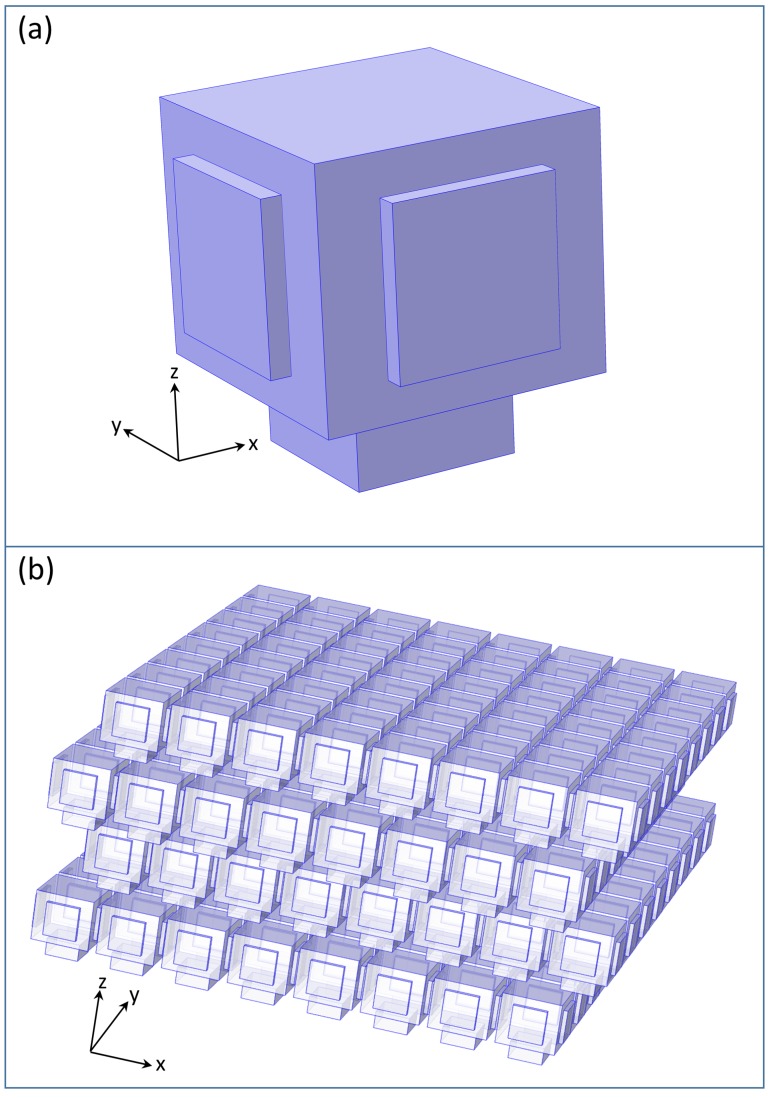
Network of the air voids in the photonic nanoarchitecture. (**a**) Unit cell, with a rectangular air void and horizontal and vertical channels. The real horizontal channels connecting the air voids are twice as long as that seen in this figure because the boundary plane of the unit cell cuts these channels in half. (**b**) The four-story structure built from 3D tiling of the space with the building block shown in Figure 3a. It is periodic in the xy directions and has four stories in the z direction. The adjacent stories are shifted by a half lattice constant in the x and y directions. An 8 × 8 cell section is shown in this figure; the real model structure is infinite in the x and y directions. The dimensions of the model are given in Figure 4 and Table 1. In the real structure, the spaces between the air voids are filled with chitin. *x* and *y* are the horizontal directions, and *z* is the vertical direction. The light is impinging from the upper (−z) direction.

**Figure 4 nanomaterials-09-00759-f004:**
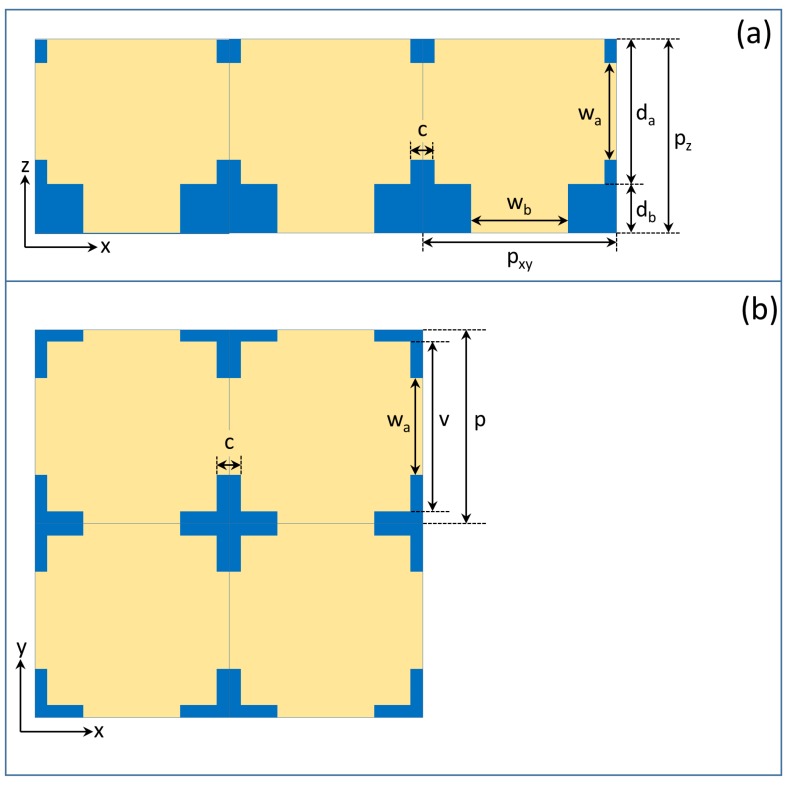
Vertical and horizontal cross-sections of the 3D model shown in Figure 3, with the characteristic sizes. Blue is chitin, and yellow is air. (**a**) The vertical cross-section of one-story, three-unit cells is shown. (**b**) The horizontal cross-section of a 2 × 2 unit cell section is displayed.

**Figure 5 nanomaterials-09-00759-f005:**
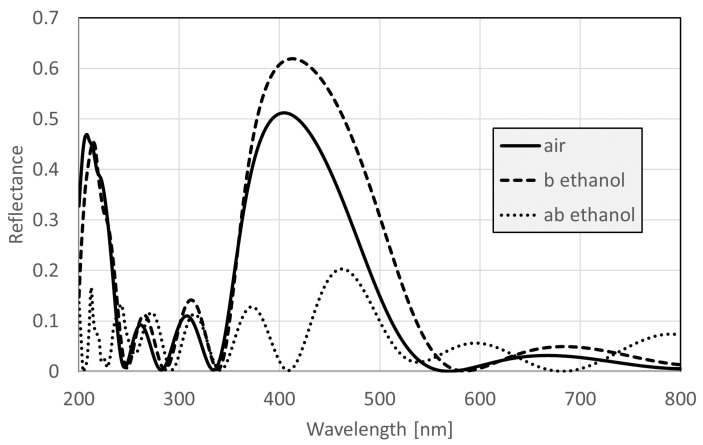
Effect of the ethanol filling of the “A” and “B” layers on the FEM calculated reflectance of one dorsal scale. Solid line: voids in both the “A” and “B” layers are filled with air. Dashed line: voids in the “B” layers are filled with ethanol. Dotted line: voids in both the “A” and “B” layers are filled with ethanol.

**Figure 6 nanomaterials-09-00759-f006:**
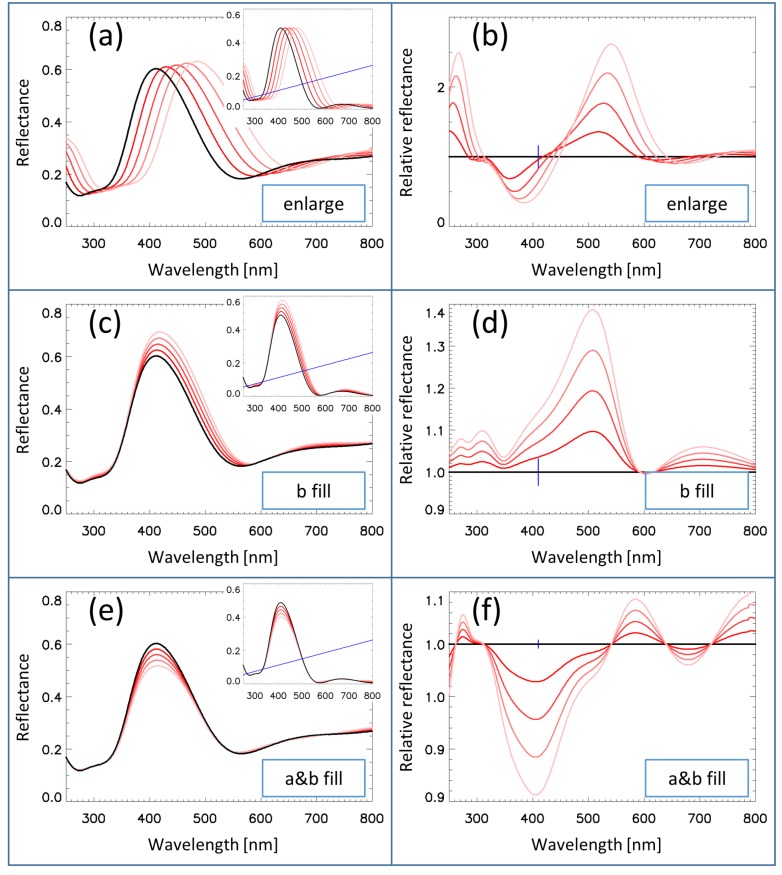
Effect of the gradual changing of different simulation parameters on the calculated dorsal reflectance spectrum of the *Albulina metallica* butterfly. (**Left**) Simulated reflectance spectra for the whole wing. (Left insets) Simulated reflectance spectra for single scales. The blue line shows the background added to these spectral curves in order to model the whole wing spectra. (**Right**) Simulated relative reflectance spectra for the whole wing. The small vertical bars denote the position of the peak wavelength of the reflectance. (**a**,**b**) Isotropic enlargement of the structure by 4%, 8%, 12%, and 16%. (**c**,**d**) Effect of the $c_{B}$ parameter for 0.2, 0.4, 0.6, and 0.8 values. (**e**,**f**) Effect of the $c_{AB}$ parameter for 0.05, 0.10, 0.15, and 0.20 values. Note the identical vertical scales on Subfigures (**a**,**c**,**e**) and the different vertical scales on Subfigures (**b**,**d**,**f**).

**Figure 7 nanomaterials-09-00759-f007:**
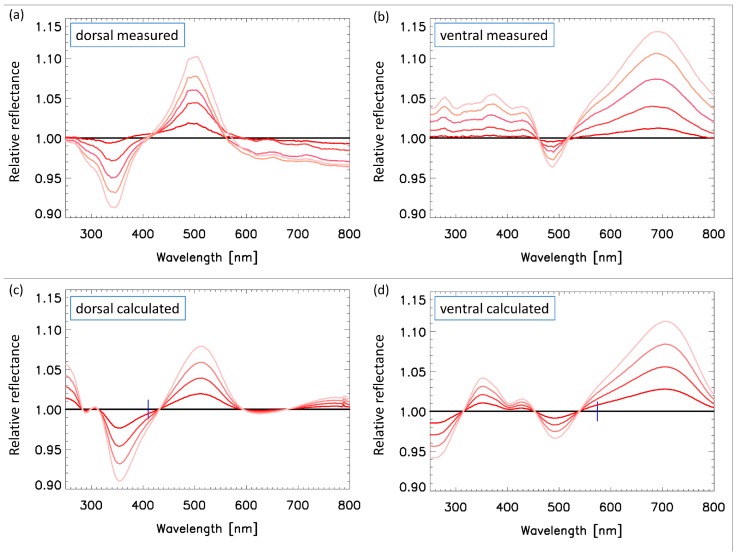
Comparison of the measured and calculated vapor-dependent dorsal and ventral relative reflectance spectra of the *Albulina metallica* butterfly. (**a**,**b**) Measured spectra for 10%, 20%, 30%, 40%, and 50% ethanol vapor concentrations. (**c**,**d**) Calculated spectra for 12.5%, 25.0%, 37.5%, and 50.0% ethanol vapor concentrations.

**Table 1 nanomaterials-09-00759-t001:** Dimensions of the 3D model. All sizes are in nm.

Quantity	Symbol	Dorsal	Ventral
Lateral periodicity	$p_{xy}$	200	260
Vertical periodicity	$p_{z}$	200	260
“A” layer thickness	$d_{a}$	150	195
“B” layer thickness	$d_{b}$	50	65
Air void width	$v_{xy}=p_{xy}-c$	175	227.5
Air void height	$v_{z}=d_{a}$	150	195
Horizontal air channel width	$w_{a}$	100	130
Vertical air channel width	$w_{b}$	100	130
Horizontal air channel length	*c*	25	32.5
Vertical air channel length	$d_{b}$	50	65

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
