# Peer review of "Modeling the Reflectance Changes Induced by Vapor Condensation in Lycaenid Butterfly Wing Scales Colored by Photonic Nanoarchitectures"

_nanomaterials, 2019, doi:10.3390/nano9050759_

Reviewer 1 Report

This paper presents a courageous investigation of the spectral changes induced by vapours in butterfly wing scales. The complex structure of the studied lycaenid scales is first delineated and then a theoretical/numerical analysis using FEM modelling is given. The paper is generally well presented and so I have only a few questions and suggestions for improvement. 

130-154           Whereas the description of the non-porous vs porous B layers is very confusing, the lines 154-158 are somewhat redundant/obvious from the figure.

245      A color is not reflected, and certainly not given by a wavelength.

288-230           This is difficult. Firm evidence for the statement that the coherence length is as short as assumed here is not provided by Ref. 38, i.e. I do not believe that reflectances should be added incoherently without extensive further argument. For the time being I like to assume that the scales act as multilayers.

300      Melanin can be eumelanin or phaeomelanin, which have different absorption spectra.

Figure 6.          The blue line should be explained as to be due to scattered/reflected light that increases with wavelength because of the decreasing absorption by melanin. It is therefore slightly unsatisfactory that the imaginary part of the refractive index of the (melanin) pigmented chitin is assumed to be wavelength-independent. (Why are all spectra red and not given different colours?)

Typography

Add space before the numbered references in the text. The positioning of the references is often awkward, for instance in lines 48, 64, 60, 66, 68. Generally, it is good practice to put the reference at the end of a sentence.

62        Delete comma

66, 67  Delete one first.

122      chitin and ethanol are not parameters and therefore should not be in italics. Idem the subscripts further on, for instance in lines 185-188 and section 3.4.

142      an > a

146      a very close value – change this into something like a value close to … or, similar values

246      To use the same letter (p) for a volume fraction (243, 244) and an optical path length is bad practice and should be avoided. Replace the first p by f, for instance (as in 279).

262      Same, to use ʋ both for volume (243) and length (262, 263) is a no no.

The references need scrutiny, for instance for insect names in italics, journal abbreviations, no capitals (Ref. 1, 27, 37…), Ref. 30 is a book and hence needs an editor and publisher.

Author Response

Response to Reviewer 1 Comments

We are grateful to Reviewer 1 for his devoted work and helpful suggestions.

Point 1: 130-154 Whereas the description of the non-porous vs porous B layers is very confusing, the lines 154-158 are somewhat redundant/obvious from the figure.

Response 1: While we agree with the Reviewer that for an experienced reader this part may seem obvious, for the benefit of a Reader not expert in the TEM filed we think that it may be useful to emphasize this. We inserted a brief explanation (with blue), stressing that the size of the smaller pores is on the same scale as the thickness of the TEM slice, hence the gray value of a given pixel on the TEM image depends on the size of the air void.

Point 2: 245 A color is not reflected, and certainly not given by a wavelength.

Response 2: The phrase “dominant reflected color” is replaced with “main visible reflectance spectral peak wavelength” in the text.

Point 3: 288-230 This is difficult. Firm evidence for the statement that the coherence length is as short as assumed here is not provided by Ref. 38, i.e. I do not believe that reflectances should be added incoherently without extensive further argument. For the time being I like to assume that the scales act as multilayers.

Response 3: We try to formulate our point more clearly. There is no disagreement between us that the structure is a multilayer structure, with layer thicknesses within the coherence length. Maybe it was not the best choice to cite the Namur paper, as indeed it is somewhat confusing. We just wanted to point out that if we separate the structure into a fraction in which the voids that are completely filled with air and another fraction in which the voids are completely filled with condensed liquid, the lateral distance of the completely filled voids will be larger than the coherence length, so that we can simply add the two contributions from the filled and from the empty fractions.

We replaced the phrase “average distance” with “average lateral distance” in the text in order to clarify our point.

Point 4: 300      Melanin can be eumelanin or phaeomelanin, which have different absorption spectra.

Response 4: We inserted a new sentence about this with a new reference (41).

Point 5: The blue line should be explained as to be due to scattered/reflected light that increases with wavelength because of the decreasing absorption by melanin. It is therefore slightly unsatisfactory that the imaginary part of the refractive index of the (melanin) pigmented chitin is assumed to be wavelength-independent.

Response 5: We inserted a new explanation, pointing out that in the linear approximation, where the linear function has a constant term and a wavelength dependent term. First we applied a constant (wavelength independent) refraction index and the wavelength dependent part of the function was approximated by a linear function.

Point 6: Why are all spectra red and not given different colours?

Response 6: These figures were originally designed keeping in mind that they have to remain understandable in a grayscale print, too. That’s why we applied different shades of red. Since the MDPI Nanomaterials is an online journal, we do not have this requirement any more. Hence, we are going to replace the different shades of gray with different colors in the production phase.

Point 7: Add space before the numbered references in the text. The positioning of the references is often awkward, for instance in lines 48, 64, 60, 66, 68. Generally, it is good practice to put the reference at the end of a sentence.

Response 7: We corrected these points.

Point 8: 62 Delete comma

Response 8: That was inserted by the language lector of the MDPI during the English editing process.

Point 9: 66, 67  Delete one first.

Response 9: We don’t find this in the text.

Point 10: 122      chitin and ethanol are not parameters and therefore should not be in italics. Idem the subscripts further on, for instance in lines 185-188 and section 3.4.

Response 10: We didn’t use any special formatting, just applied the macro package of MDPI and that formats variable names such a way.

Point 11: 142      an > a

Response 11: Corrected

Point 12: 146 a very close value – change this into something like a value close to … or, similar values

Response 12: Corrected

Point 13: 246 To use the same letter (p) for a volume fraction (243, 244) and an optical path length is bad practice and should be avoided. Replace the first p by f, for instance (as in 279).

Response 13: Thank you for noticing this. We left “p” for the optical path length and used “P” for the porosity. We use “f” for the filling factor, which is the ratio of liquid filled volume to the total air volume.

Point 14: 262 Same, to use ʋ both for volume (243) and length (262, 263) is a no no.

Response 14: Thank you for noticing this. We left “v” for the size of the air void and introduced “V” for the volume. And…there was a third “v” in the paper, the ethanol vapour concentration. We replaced that with “x”.

Point 15: The references need scrutiny, for instance for insect names in italics, journal abbreviations, no capitals (Ref. 1, 27, 37…)

Response 15: Corrected

Reviewer 2 Report

This is a well written paper presenting results of matching a parametric simulation model of butterfly wings to measured spectra. The results provide new insight into the workings of butterfly wings as chemical sensors. The manuscript is publishable in its present form. 

Author Response

We are grateful for the Referee for evaluating our work.

Reviewer 3 Report

Mark and colleagues in their paper present a theoretical study concerning the gas/vapour response of lycaenid butterflies and show that a rather simple model of stacked 3D cubes appoximates the optical response. The paper reads ok, but could be majorly improved to clarify the findings and makes it accessible to a broader audience. Let me detail some points below.

The manuscript is very long and full of detail that no reader will ever logically need. I would suggest to move large parts of this to a Supplementary Information file and concentrate the main manuscript onto the main message, namely that DSA works and that it can explain the response. This way, a clear message will be shown and the results can be discussed a bit more broadly.

Concerning the discussion, I miss a generalization of the results. How does chitin swelling contribute, is the degree you predict reasonable and can these factors be used in other ways to design gas sensors with a sensible response? I would much enjoy reading this, than the current discussion that is more repeating the results of the paper.

Next to shortening/focussing, the paper could benefit from another read of  a friendly speaker. It’s generally quite well written, but some issues with time and presentation persists. Already in the summary it is for example not clear what has been done in this paper and what has been done by others before.

The citations need to checked (some miss journal names, some are misspelled, etc.), but generally a slightly broader citation style to others work would help giving this work more impact.

The methods need more detail. What is the slight modification in line 94? When does the reflectance change by 4.5 nm in line 103? Just to name a few. Also the approximation of a imaginary RI for the chitin is somewhat too simplified (see Stavengas recent work on RI and pigments) that at least a comment on the validity over the entire wavelength range is needed.

I hope these comments are helpful for a revision.

Author Response

Response to Reviewer 3 Comments

We wish to thank to Reviewer for his valuable recommendations and for his work in helping us improve the manuscript. Also, we are thankful for his pointing out that the chitin swelling could be the subject of a more extended paper, which we will consider in the future.

Point 1:

Mark and colleagues in their paper present a theoretical study concerning the gas/vapour response of lycaenid butterflies and show that a rather simple model of stacked 3D cubes appoximates the optical response. The paper reads ok, but could be majorly improved to clarify the findings and makes it accessible to a broader audience. Let me detail some points below.

The manuscript is very long and full of detail that no reader will ever logically need. I would suggest to move large parts of this to a Supplementary Information file and concentrate the main manuscript onto the main message, namely that DSA works and that it can explain the response. This way, a clear message will be shown and the results can be discussed a bit more broadly.

Response 1:

While we understand the point of view of the Reviewer, unfortunately, we cannot fully agree with this recommendation. The paper is intended for a broad audience ranging from nanoscale physics, to materials science, biology and sensorics, so we feel that the details presented in the paper are needed as researchers in the different fields have different backgrounds. As the other two Reviewers did not recommended such a drastic modification of the structure of the paper, we think that the present structure of the paper is justified.

Point 2:

Concerning the discussion, I miss a generalization of the results. How does chitin swelling contribute, is the degree you predict reasonable and can these factors be used in other ways to design gas sensors with a sensible response? I would much enjoy reading this, than the current discussion that is more repeating the results of the paper.

Response 2:

We thank to the Reviewer for this recommendation and in subsequent paper we intend to go into more detail in this matter of the chitin swelling. However, the present paper is not intended to study the effects of chitin swelling per se, it is the modeling of the experimentally observed behavior of the butterfly wing which is in the focus of our paper.

Point 3:

Next to shortening/focussing, the paper could benefit from another read of a friendly speaker. It’s generally quite well written, but some issues with time and presentation persists.

Response 3: The paper has been language edited at the MDPI.

Point 4: What is the slight modification in line 94?

Response 4: Clarified in the text.

Point 5: When does the reflectance change by 4.5 nm in line 103?

Response 5: Clarified in the text.

Point 6: Also the approximation of a imaginary RI for the chitin is somewhat too simplified (see Stavengas recent work on RI and pigments) that at least a comment on the validity over the entire wavelength range is needed.

Response 6: We included references for two Stavenga papers and included a pair of sentences to clarify this part.

Point 7: I hope these comments are helpful for a revision.

Response 7: Thank you very much indeed.

Round  2

Reviewer 3 Report

I understand why the authors chose not to shorten the paper, but I'm honestly not too happy with their choice. The ms is just a lot to read and shortening it to the main points by moving unnecessary noise to an SI would help making this paper digestable and giving it the impact it deserves. I just fear that the current version is containing too much information to be accesible (and the discussion on disentangling swelling and filling is still missing). I hoped my comments would aid the authors to that point, but see that they decided not to follow it. No further comments from me.

Author Response

We are very much indebted to the 3rd Reviewer, for replaying so fast.

We are glad that the reviewer understands that we decided to keep the present format of our paper. As we already pointed out in our previous reply to the Reviewer, the paper is intended for a broad range of readers therefore, the problem has to be presented from several angles. Some of which may seem “technicalities” for a reader with a certain expertise. As the other two Reviewers did not ask for such major revisions in the structure of the manuscript, we believe that the structure of our manuscript is justified by the complexity of the topic.